# Metal Oxide Nanoparticles and Nanotubes: Ultrasmall Nanostructures to Engineer Antibacterial and Improved Dental Adhesives and Composites

**DOI:** 10.3390/bioengineering8100146

**Published:** 2021-10-19

**Authors:** Abdulrahman A. Balhaddad, Isadora M. Garcia, Lamia Mokeem, Rashed Alsahafi, Fabrício Mezzomo Collares, Mary Anne Sampaio de Melo

**Affiliations:** 1Department of Restorative Dental Sciences, College of Dentistry, Imam Abdulrahman Bin Faisal University, Dammam 34212, Saudi Arabia; 2Program in Dental Biomedical Science, School of Dentistry, University of Maryland, Baltimore, MD 21201, USA; lsmokeem@umaryland.edu; 3Dental Materials Department, School of Dentistry, Federal University of Rio Grande do Sul, Porto Alegre 90035-003, RS, Brazil; isadora.mgarcia@hotmail.com (I.M.G.); fabricio.collares@ufrgs.br (F.M.C.); 4Department of Restorative Dental Sciences, College of Dentistry, Umm Al-Qura University, Makkah 24381, Saudi Arabia; rashed.alsahafi@umaryland.edu; 5Operative Dentistry Division, General Dentistry Department University of Maryland School of Dentistry, Baltimore, MD 21201, USA

**Keywords:** antibacterial, adhesives, bioactivities, nanoparticles, resin composite

## Abstract

Advances in nanotechnology have unlocked exclusive and relevant capabilities that are being applied to develop new dental restorative materials. Metal oxide nanoparticles and nanotubes perform functions relevant to a range of dental purposes beyond the traditional role of filler reinforcement—they can release ions from their inorganic compounds damaging oral pathogens, deliver calcium phosphate compounds, provide contrast during imaging, protect dental tissues during a bacterial acid attack, and improve the mineral content of the bonding interface. These capabilities make metal oxide nanoparticles and nanotubes useful for dental adhesives and composites, as these materials are the most used restorative materials in daily dental practice for tooth restorations. Secondary caries and material fractures have been recognized as the most common routes for the failure of composite restorations and bonding interface in the clinical setting. This review covers the significant capabilities of metal oxide nanoparticles and nanotubes incorporated into dental adhesives and composites, focusing on the novel benefits of antibacterial properties and how they relate to their translational applications in restorative dentistry. We pay close attention to how the development of contemporary antibacterial dental materials requires extensive interdisciplinary collaboration to accomplish particular and complex biological tasks to tackle secondary caries. We complement our discussion of dental adhesives and composites containing metal oxide nanoparticles and nanotubes with considerations needed for clinical application. We anticipate that readers will gain a complete picture of the expansive possibilities of using metal oxide nanoparticles and nanotubes to develop new dental materials and inspire further interdisciplinary development in this area.

## 1. Introduction

Dental caries has been classified as the most common oral disease as it affects around 60–90% of adolescents and 100% of grownups around the world [1,2]. Dentists use a variety of restorative materials to treat dental decay cavities. Currently, dental restorations, created with dental adhesives and composites, are the most used materials for this purpose [3]. However, the inherent composition of dental adhesives and composites has a high affinity for dental plaque (biofilms) accumulation, leading to the recurrence of cavitation (secondary caries) after the tooth has been restored with these materials [4]. Additionally, resin-based restorative materials are highly susceptible to degradation, which may compromise the strength of such dental restorations over time [5]. As a result, secondary caries and restoration fractures are the most common reasons for the premature failure of dental restorations created with dental adhesives and composites [6] (Figure 1). The restorations often need to be replaced within 5–7 years, leading to more extensive restorations weakening the tooth [7,8,9,10].

New investigations on restorative materials address the use of metal oxide nanoparticles and nanotubes to impart antibacterial properties and enhance the mechanical properties of materials, such as dental bonding agents, cements, and resin composites. Up to date, nanostructures present two pathways to impart an antibacterial effect into resin-based materials: the release of ions that interact with oral pathogens inhibiting bacterial growth and the release of ions that decrease tooth demineralization [11,12,13]. However, when combining an antibacterial agent with resin-based materials, several factors must be considered to avoid turning down the materials’ mechanical properties or increasing their toxicity [14]. 

As the range of applications of metal oxide nanoparticles and nanotubes has dramatically increased over time, they have unsurprisingly reached the dental field. Here, we primarily focus on the contributions of metal oxide nanoparticles and nanotubes to restorative dentistry, emphasizing the capability to convey the dental adhesives and composites with antibacterial activity. We begin with a discussion of the ongoing burden of the premature failure of composite restorations. From there, we provide a general discussion on the use of nanotechnology to improve the field of restorative dentistry. We then review the cutting-edge investigations of metal oxide nanoparticles and nanotubes incorporated into dental materials. The following Section summarizes key considerations for designing new dental materials meant to reduce premature failure. Finally, we round out our discussion on these nanostructures with a perspective of future directions and challenges for the clinical application. Overall, we anticipate that readers will gain a greater perspective on the range of possibilities available for metal oxide nanoparticles and nanotubes and the need for a vibrant interdisciplinary collaboration to fully translate this potential into new dental materials.

## 2. How Long Do Composite Restorations Last?

One of the objectives of restorative treatment is the faithful reproduction of the characteristics of natural teeth, such as the color and shape. The search for materials for direct restoration with properties similar to the hard dental tissues led to the development of composite resins. With the advent of adhesion to hard dental tissues, the resin composites were also improved with adhesive systems.

With the introduction of enamel acid etching by Buonocore in 1955 [15], there was a paradigm shift in restorative dentistry. Previously, the restorative material defined the cavity preparation, especially using dental amalgam. Currently, it is possible to work with a minimally invasive perspective using adhesive systems and composite resins. Based on that, the tooth cavity can be prepared considering only the irreversibly compromised dental tissue since the material will be bonded to the sound tooth, and additional retention for the materials is not required. In addition, the resin composites can reinforce the remaining tooth structure, provide a greater possibility of restoration repair, the expansion of aesthetic options, and expansion of clinical indications.

Regarding the longevity of composite resin treatments on posterior teeth, a review selected clinical trials that investigated restorations during at least five years of follow-up, published between 1996 and 2011. A total of 90% of clinical studies indicated an annual failure rate between 1% and 3% for Class I and II, depending on factors such as the type and location of the tooth, operator, and socioeconomic, demographic, and behavioral elements. The main reasons for long-term failure were recurrent caries and fractures, related to caries risk and parafunctional habits, respectively [16].

Another article on the longevity of posterior teeth restorations performed a systematic review with a meta-analysis with longitudinal studies with at least five years of follow-up between 1990 and 2013 [17]. The development of caries as a cause of failure of restorations, specifically composite resin, is the essential factor starting five years after restoration placement. In this scenario, patients at high risk of caries had an annual failure rate of 4.6% in ten years, while those at low risk of caries showed a value of 1.6% in the same period. Thus, the annual failure rate of patients at high risk of caries was almost three times higher than those at a low caries risk. This same study showed that, while low-risk patients had about a 60% survival of restorations, those at high risk of caries had a survival of only 35% in 20 years [17].

A 29-year follow-up study revealed a cumulative survival rate of 91.7% at six years, 81.6% at 12 years, and 71.4% at 19 years. The average annual failure rate was 1.92% [18]. Another study with an extended follow-up showed a similar failure rate and survival time [19]. Class II restorations were evaluated, totaling an annual failure rate of 1.6%. In this context, 54% of failures were caused by recurrent caries. While low-risk patients showed a probability of the survival of restorations of 66%, patients at high risk of caries showed a probability of only 35% [19]. 

Therefore, the survival of restorations depends not only on the material, but also on factors related to the patient and the operator. As caries is a biofilm-sugar-dependent disease, with social class, income, knowledge, education, and behavior as factors that modify the severity of the disease, clearly patient-centered clinical approaches should always focus on the maintenance and greater longevity of restorations [19]. 

As for the longevity of treatments with composite resin on anterior teeth, the results are also satisfactory. A retrospective longitudinal clinical study that considered only permanent tooth restorations performed by general practitioners showed an annual failure rate of 4.6% in ten years of follow-up for approximately 70,000 anterior tooth restorations evaluated [20].

In general, composite resins and dental adhesives support a long-lasting and reliable treatment, the leading choice for anterior and posterior tooth restoration. However, factors related to patients and dentists influence the outcome. Therefore, the development of restorative materials with improved therapeutic properties, such as antimicrobial activity and bioactivity, has been researched. Thus, it is expected that these materials will help maintain treatments, especially for patients in a risk group.

Several approaches were followed to integrate bioactivity into resin-based restorative materials, including contact-killing and protein-repelling agents, inorganic metallic particles, and ion-releasing materials [3,4,21,22,23]. Contact-killing Quaternary ammonium methacrylates (QAMs) were copolymerized in resins to accomplish antibacterial activities [13,24,25]. With the increase in the alkyl chain length (CL) in its chemical structure, the antibacterial capability of QAMs could be increased [21]. A salivary proteins coating on the surface of the resin provides the medium for bacterial adhesion to the composite [26,27]. Thus, it is important to repel the salivary proteins from the composite surface. 2-methacryloyoxyethyl phosphorylcholine (MPC) is a common biopolymer with phospholipid polar groups [28,29]. Resins containing MPC had favorable protein-repellent properties, which enhanced the inhibition efficacy of the resins against cariogenic biofilms [30,31]. 

Release-based antibacterial materials such as silver nanoparticles (Ag) show the ability to retain a long-lasting antimicrobial effect [32,33] and have less bacterial resistance than antibiotics [34]. Several studies had integrated nanoparticles of silver (Nag) into dental composites, which effectively inhibited the growth of *Streptococcus mutans* [35]. For remineralization purposes, bioactive composites enclose nanoparticles of amorphous calcium phosphate (NACP), which exhibited long-lasting calcium (Ca) and phosphate (P) ion releases [36,37,38]. The resins containing NACP have an acid-neutralization capability to shield the tooth surfaces from acid challenges. In addition, the released ions stimulated tooth remineralization, resulting in mineral regeneration in the tooth lesions [36,37,38]. It is essential to mention that some of these antibacterial agents showed limited long-term effects. In contrast, others showed a satisfactory ability to reduce tooth demineralization, inhibit artificial secondary caries, and improve long-term bond strength. However, most of the available studies lack the clinical evaluation for those combined materials. Therefore, it is highly recommended to continue developing those antibacterial materials and investigate their performance clinically. 

In addition to the fact that composite resins can be modified to provide bioactivity and antibacterial activity, adhesive systems also play an essential role in maintaining composite resin restorations. Therefore, these materials are also being modified. The most extensive study evaluating the clinical efficacy of adhesive systems was published in 2014. The study is a systematic review of papers in which adhesive systems were used in randomized clinical trials in non-carious cervical lesions from 1950 to 2013. The authors divided the adhesive systems into three-step conventional, two-step conventional, two-step self-etching, and one-step self-etching. The self-etching products were further divided according to pH into light and moderately strong, with pH ≥ 1.5; strong, with pH < 1.5 [39]. The lowest annual failure rate was observed for the two-step self-etching adhesive system with a moderate pH (2.5 (±1.5)), followed by the conventional three-step adhesive system (3.1 (±2)) and the one-step self-etching adhesive system with a moderate pH (3.6 (±4.3)), with no statistically significant difference between these systems. Higher annual failure rates were observed for strong one-step self-etching adhesive systems (5.4 (4.8)), two-step conventional adhesive system (5.8 (4.9)), and strong two-step self-etching adhesives (8.4 (7.9)), with no statistical difference between these three adhesive systems [39]. 

Regarding universal adhesive systems, the follow-up time is short compared to other systems. However, there are good results so far. The study with the most extended follow-up randomized clinical trial dates back to 2020 [40], and analyzed the survival of restorations in non-carious cervical lesions using a universal adhesive and different acid etching strategies. After five years, retention/fracture rates were between 81.4 and 93%. The purpose of reporting these data is to highlight the importance of dental adhesives in improving the survival rate of resin composite restorations. As a result, several investigations were reported to modify dental adhesives to enhance their performance and impart bioactivity.

## 3. Nanotechnology in Dentistry

Nanotechnology is defined as the presentation of scientific knowledge to employ and control matter primarily in the nanoscale to obtain the benefit of size and structure-dependent properties and phenomena distinct from those connected with individual atoms, molecules, or extrapolation from bigger sizes of the same material [41]. Recently, this technology has been utilized in medicine. It is known as nanomedicine, which has broadened the horizons toward new pathways in human diseases diagnosis and treatment [42]. The dental field was also included in this era using nanoparticles [43]. They were used in dental fillings, dental implants, caries prevention, and teeth whitening [44]. 

Nanoparticles were inserted into many products to improve the properties of resin-based composites, such as polishability and gloss stability. The size of these nanoparticles ranges from 1 nm to 100 nm [45]. As a result, they were promising in antibacterial therapies due to their high chemical reactivity, unique ultra-small size, and a large surface area to mass ratio [46]. In addition to the elevated charge density, this large surface area has assisted the nanoparticles in interacting with the negatively charged surface of bacterial cells to a superior extent, boosting the antimicrobial activity [47]. Additionally, combining the nanoparticles with polymers or coating them into biomaterial surfaces increased their antimicrobial properties [46]. 

Nanoparticles investigated for dental applications were divided into metal and organic particles. They showed a broad spectrum of bactericidal properties that led to expansion in their use [48]. As a result, a plethora of investigations was designed for this purpose. Studies had divided the antibacterial mechanisms of nanoparticles into (1) cell lysis due to the interaction with the peptidoglycan cell wall and membrane, (2) disturbance in protein synthesis due to bacterial interaction with proteins, and, lastly, (3) the prevention of DNA replication when the bacterial cytoplasms interact with DNA [49,50,51]. Moreover, the use of nanoparticles was implemented to improve the mechanical and physical properties of restorative materials, such as restorations’ strength and optical properties. 

## 4. The Use of Metallic Oxide Particles in Restorative Dentistry

Metal oxide particles are widely used in industry and medical fields. These particles are mainly manipulated in size, shape, crystallinity, and functionality to obtain desirable properties in the synthesized material [52]. Moreover, despite the size scale, the high density of corners and edges of these metal oxides provides a wide range of desired chemical and physical properties [53]. In restorative dentistry, metallic oxide particles have gained much attention, as they can be used to improve the materials’ properties and bioactivity. Table 1 summarizes the most recent findings related to the use of different metal oxides in restorative dentistry. 

### 4.1. Titanium Dioxide (TiO_2_)

Nanostructures of titanium dioxide (TiO_2_) are one of the most abundant emerged elements in numerous medical and nonmedical technology fields [73]. TiO_2_ is a poorly soluble white material with different medical, pharmaceutical, and cosmetics applications [74,75,76]. Two crystalline structures form TiO_2_, rutile and anatase. The rutile form of TiO_2_ is less active than the anatase form regarding cytotoxic properties and photo-catalytic activity [76]. Mixed polymorphs of TiO_2_ have demonstrated improved outcomes for biomedical use than the use of one crystal. Titanium dioxide nanostructures are cost-effective non-toxic substances with a wide range of potential uses [76]. Nanostructured TiO_2_ was approved for use in food and drug-related products by the American Food and Drug Administration (FDA) in 1966 [77]. 

In dentistry, caries is one of the most widespread chronic diseases worldwide. Dental caries affects the oral and general health and creates considerable public healthcare and economic burdens [78]. To overcome this difficulty, TiO_2_ nanoparticles have been used in dental resin. According to previously published studies, the use of TiO_2_ nano in the resin matrix has yielded a potent antibacterial result in composites, sealants, bases, liners, adhesive, and cement [73,79]. In research conducted by Sodagar et al. [54], they evaluated the antimicrobial and mechanical properties of resin campsites modified by adding different concentrations of nanoparticles of TiO_2_ (0%, 1%, 5%, and 10%) to be used as an adhesive. They found that all concentrations of TiO_2_ nanoparticles significantly reduced the colony counts of *S. mutans* and *Streptococcus sanguinis*.

Moreover, all experimental groups had a significant effect on the formation and extension of the inhibition zone. Thus, the shear bond strength of composite containing 1% and 5% nanoparticles was still in an acceptable range. On the other hand, the composite with 10% nanoparticles showed significantly lower results than other groups. 

TiO_2_ nanoparticles are extremely small particles that are difficult to disperse in an organic solvent and can agglomerate easily in practical uses [80]. To overcome this drawback, Xia et al. [55] modified TiO_2_ nanoparticles with organosilane allyltriethoxysilane (ATES) and blended them into dental resin-based composites. They found that surface coatings with the organosilane ATES improved the linkage and dispersion of TiO_2_ nanoparticles within a resin-based composite. Moreover, the microhardness and flexural strength of the dental resin-based composite were improved by adding the modified nanoparticles compared to the control. 

Another promising feature of TiO_2_ nanoparticles is the unique photo-catalytic activities (PCAs). PCAs make TiO_2_ nanoparticles attractive to be used in high-performance dental resin-based materials [1,2,3,4]. TiO_2_ nanoparticles use energy from light irradiation to generate electrons and convert water and oxygen into powerful free radicals and oxidation agents [81]. Sun et al. [56] incorporated acid-modified nanoparticles of TiO_2_ into a dental adhesive at a mixture of bis-phenol-A-dimethacrylate (BisGMA) and triethylene glycol dimethacrylate (TEGDMA) (mass ratio 1:1) at mass fractions (0%, 0.02%, 0.03%, 0.05%, 0.06%, 0.08% and 0.5%). The distribution of the acid-modified TiO_2_ nanoparticles in ethanol was found to agglomerate with relatively narrow sizes. By adding a small amount of acid-modified TiO_2_ nanoparticles, the outcome of the resin mixture was enhanced considerably. The hardness and elastic modulus of the BisGMA and TEGDMA resins were improved significantly by adding a 0.06% mass fraction of modified TiO_2_ nanoparticles by around 48% for the elastic modulus and more than double for the hardness compared to nanoparticle-free resin. The degree of resin conversion was improved by approximately 5% by adding a 0.08% mass fraction of nanoparticles. The mean shear bond strength was improved by about 30% when a 0.1% mass fraction of nanoparticles was incorporated. 

Another study by Sun et al. [57] found that the functionalization of TiO_2_ nanoparticles improved photo-catalytic activities in producing free radicals under visible light irradiation. TiO_2_ nanoparticles at a tiny fraction in the resins can achieve a dramatic performance enhancement. Moreover, through the functional groups, including carboxyl and hydroxyl groups, their uses can be tuned.

### 4.2. Zinc Oxide (ZnO)

Zinc oxide (ZnO) is a white inorganic compound insoluble in water with a ZnO formula [82]. In dentistry, ZnO macro and micro-sized particles have been examined as an inorganic filler in restorative materials. Composites, adhesive resins, sealers, and cement are among the materials studied by ZnO [59,83]. Smaller particles of ZnO nanoparticles demonstrated a superior antimicrobial activity than larger particles against both Gram-negative and Gram-positive bacteria [84]. In addition, nano-sized particles have a high surface area to volume ratio; therefore, they have a higher percentage of atoms on the material’s surface, leading to an increased surface reactivity [85]. Nano-sized ZnO particles through bioactivity can enable mineral growth and negatively affect bacteria growth [85,86,87]. Consequently, their biological properties could be mainly related to their higher reactivity and low dimensionality. 

In research conducted by Garcia et al. [59], nano-sized ZnO was incorporated at 0, 2.5, 5, and 7.5 wt.% in an experimental dental adhesive (Figure 2A). They found that ZnO nanoparticles at 7.5% demonstrated a substantial bacterial reduction against a 48h mature saliva-derived oral microcosm biofilm. The chemical and mechanical properties of the dental adhesive were affected by the addition of 7.5 wt.% of ZnO nanoparticles. Nevertheless, the flexural strength was within the ISO recommended range, and the degree of conversion was comparable to the reported values of commercial adhesives [59]. Another study evaluated the mechanical properties and the antibacterial activity of resin composites containing 0–5 wt.% ZnO nanoparticles against *S. mutans*. It found that the addition of all ZnO nanoparticles with a concentration up to 5 wt.% into a flowable resin composite would significantly inhibit the growth of *S. mutans*. However, only the incorporation of the nanoparticles up to 1 wt.% would not adversely affect the mechanical properties of the composite [58]. Recently, different methods allowed the cost-effective synthesis of different nanostructures with various morphologies of ZnO nanoparticles [60,88]. Grown ZnO nanoparticle structures, such as needles, tetrapods, etc., have exhibited an exciting step to achieve better dental materials. Collares et al. [88] evaluated the effect of needle-like zinc oxide nanostructures (ZnO-NN) at 0, 20, 30, and 40 wt.% on the antibacterial, physical, and chemical properties of experimental methacrylate-based dental sealers. They found that all concentrations of ZnO-NN (20, 30, or 40 wt.%) decreased the E. faecalis growth without a significant detrimental effect on the physical and chemical functionality of the material. Another study investigated a composite resin’s antibacterial and mechanical properties containing a 0, 3, 5, 10 wt.% of tetrapod-like zinc oxide whisker (T-ZnOw). The results revealed that incorporating T-ZnOw fillers improved both the antibacterial activity and the mechanical properties of the experimental composite resin. Furthermore, incorporating 5% wt.% of T-ZnOw into the resin was optimum for enhancing mechanical properties, antibacterial action, and long-term antibacterial efficiency [60].

### 4.3. Copper Oxide Nanoparticles

Copper is a primary trace component in living organisms and is found in more than 30 kinds of proteins. Copper oxide (CuO) is a monoclinic semiconducting inorganic compound and represents the simplest member of the family of copper. CuO exhibits a range of potentially useful physical properties. CuO crystal structures provide beneficial photo-catalytic properties as well as photoconductive functionalities through their narrow band-gap. In addition, copper ions are antimicrobial by generating reactive hydroxyl free radicals and reducing sulfhydryls within cells [78,79]. Previous studies demonstrated that copper and copper alloy could kill those bacteria, yeasts, and viruses through direct contact with the material surface. Ren et al. investigated CuO nanoparticles’ antimicrobial properties and found that populations of Gram-negative (×3 strains) organisms and Gram-positive (×4 strains) organisms were reduced by 68% and 65%, respectively, in the presence of 1000 ug/mL nano CuO using time-kill assays [89]. Another study by Zajdowicz et al. investigated in vitro biofilm formation on the surface of novel copper(I)-catalyzed azide-alkyne cycloaddition (CuAAC)-based resins and found that CuAAC resins and CuAAC-based microfilled composites significantly (*p* < 0.05) reduce *S. mutans* biofilm formation by around 1–2-log compared to BisGMA-based polymers. Moreover, CuAAC-based resins have superior mechanical properties and reduced shrinkage stress [61]. 

CuAAC crosslinked networks could provide resin-based materials enhanced strength and low polymerization shrinkage stress with quantitative conversion, which could eliminate difficulties associated with BisGMA-based dental resins [62]. In one study, the flexural strength, flexural modulus, flexural toughness, and polymerization shrinkage stress of photo-curable CuAAC resin-based composites with varying filler loadings were tested and compared to a conventional BisGMA/TEGDMA-based composite. It was found that CuAAC composites with 60 wt.% microfiller generated an equivalent flexural strength of 107 ± 9 MPa, a more than twice equivalent flexural modulus of 6.1 ± 0.7 GPa, lower shrinkage stress of 0.43 ± 0.01 MPa, and more than ten times greater energy absorption of 10 ± 1 MJ m^−3^ when strained to 11% compared to BisGMA-based composites at the same filler level. Moreover, the photo-CuAAC polymerization of composites containing between 0 and 60 wt.% microfiller achieved a comparable conversion to BisGMA-based composites [90]. Another study concludes that photo-CuAAC polymerizations can form tough, glassy, low-stress homogeneous glassy crosslinked networks, achieving a high to complete conversion at an ambient temperature upon light irradiation. CuAAC polymers considerably reduced polymerization shrinkage stress and exhibited a dramatic ability to absorb energy without fracturing. Though photo-CuAAC polymerization has a slower polymerization rate compared to methacrylate-based free-radical photopolymerization, a comparable volumetric shrinkage and flexural modulus could be achieved [62].

### 4.4. Iron Oxide (Fe_2_O_3_)

Iron oxide (Fe_2_O_3_) particles have gained much attention recently as a potential material to carry and guide therapeutic agents. Several investigations have demonstrated the applicability of Fe_2_O_3_ in tissue engineering [91], cancer therapy [92], and imaging [93]. In restorative dentistry, Fe_2_O_3_ nanoparticles (Figure 2B) were used to improve the bonding strength of dental adhesives [63,64]. In one study, a dental adhesive containing 66.66 wt.% of bisphenol A glycidyl dimethacrylate (BisGMA) and 33.33 wt.% of 2-hydroxyethyl methacrylate (HEMA) as a resin matrix was modified to contain 0.0195%, 0.039%, 0.0781%, 0.1563%, 0.3125%, and 0.625% of Fe_2_O_3_ nanoparticles by weight [63]. It was found that incorporating 0.0781 and 0.1563 wt.% of Fe_2_O_3_ nanoparticles significantly improved the µ-tensile bond strength when the dental adhesives were subjected to a magnetic field with a degree of conversion and ultimate tensile strength values comparable to the control. Even when a pulpal pressure simulation model was used, the bond strength was improved by more than 15% in the Fe_2_O_3_ adhesive. The scanning electron microscopy analysis showed that the resin tags in the Fe_2_O_3_ adhesive were longer and higher in quantity than their control counterparts. The designed magnetic adhesives had shown good biocompatibility when they were exposed to human keratinocytes. Such findings may present a new pathway to improve the bonding durability of dental adhesives [63].

The same concept was applied in another study, where the incorporation of Fe_2_O_3_ nanoparticles and the magnetic force improved the dentin shear bond strength by 59% compared to the control [64]. Moreover, an antibacterial monomer and nano-sized amorphous calcium phosphate (NACP) fillers were added to the magnetic adhesive to impart antibacterial and remineralization properties. The designed antibacterial, remineralizing, and magnetic adhesives significantly reduced the growth and activities of saliva-derived biofilms [64]. The findings suggest using more than one agent to impart different characteristics in the designed restorative material. A further analysis of magnetic nanoparticles in dental adhesive may investigate the bonding durability after thermocycling or water aging. 

### 4.5. Cerium Oxide (CeO_2_)

Cerium is a rare metal that belongs to a lanthanide group with an atomic number of 58 and several biological and biomedical applications [94]. Cerium oxide (CeO_2_) nanoparticles have shown antibacterial and anti-inflammatory properties due to their capability in regenerating ROS and inducing radical scavenging and regeneration mechanisms [95,96]. In dentistry, CeO_2_ has been used in ceramic materials due to its ability to mimic the tooth enamel’s fluorescence [97]. Furthermore, CeO_2_ has a high atomic number, allowing good attenuation when the particles are subjected to dental X-ray [98]. Therefore, these particles have been suggested to improve the radiopacity of dental adhesives to facilitate the differentiation between the adhesive layer and recurrent caries (Figure 3). 

In one study, the incorporation of CeO_2_ micro-particles ranging from 0.36 to 5.76 vol.% into a dental adhesive was attempted [65]. The parental dental adhesive contained 50 wt.% bisphenol A glycol dimethacrylate (BisGMA), 25 wt.% triethylene glycol dimethacrylate (TEGDMA), and 25 wt.% 2-hydroxyethyl methacrylate (HEMA) as a resin matrix. Then, CeO_2_ particles were added at 0.36, 0.72, 1.44, 2.88, 4.32, and 5.76 vol.%. It was found that increasing the CeO_2_ concentration was associated with an increased radiopacity (Figure 3). However, the degree of conversion was reduced as the concentration increased, but the values were clinically acceptable. An amount of 1.44 vol.% of CeO_2_ was associated with the most appropriate radiopacity and degree of conversion values in the other groups [65]. A future investigation may explore the antibacterial properties of CeO_2_ particles and their influence on other mechanical and physical properties. 

### 4.6. Tantalum Oxide (Ta_2_O_5_)

Due to its antibacterial properties and biocompatibility, tantalum has adverse uses in medicine such as spine and foot, and ankle surgeries [99,100,101]. In addition, tantalum demonstrates the capability to deposit apatite over its surface, allowing the growth of hydroxyapatite and osteoblasts [102]. In dentistry, metal alloys containing tantalum were designed to improve the fracture toughness of the designed materials, such as dental implants [66,67]. Moreover, tantalum, as an opaque material, has been suggested to improve the radiopacity of dental adhesives. In one study, tantalum oxide (Ta_2_O_5_) particles were incorporated into a dental adhesive at 1, 2, 5, and 10 wt.% [68]. The parental adhesive contained 50 wt.% Bis-GMA, 25 wt.% TEGDMA, and 25 wt.% HEMA. It was found that dental adhesives containing 5 and 10 wt.% of Ta_2_O_5_ were associated with a significant increase in their radiopacity without compromising the adhesive’s degree of conversion and microhardness [68]. As in other metallic oxide particles, more studies are required to assess the antibacterial and bioactivity of Ta_2_O_5_ particles. In another investigation, tantalum oxide quantum dots (Ta_2_O_5_QDs) incorporated into dental adhesives were found to be very effective in reducing the growth of an *S. mutans* biofilm [69].

### 4.7. Niobium Pentoxide (Nb_2_O_5_)

Niobium pentoxide (Nb_2_O_5_) is a biocompatible material that has been used in several studies to improve alloys’ biocompatibility and corrosion resistance [103,104]. Moreover, Nb_2_O_5_ presents bioactivity as it was shown to improve osteoblast-like cells adhesion to implants containing Nb_2_O_5_ [105]. Similar to CeO_2_ and Ta_2_O_5_, Nb_2_O_5_ particles were incorporated into different restorative materials to improve their radiopacity. In one study, 5, 10, and 20 wt.% of Nb_2_O_5_ particles were added into a dental adhesive composed of 50 wt.% Bis-GMA, 25 wt.% TEGDMA, and 25 wt.% HEMA [70]. Their radiopacity values were increased as the concentration of Nb_2_O_5_ increased without compromising the polymerization features of the adhesive. Moreover, adding 20 wt.% of Nb_2_O_5_ significantly improved the microhardness of the material. 

In another investigation, Nb_2_O_5_ particles were added into three commercially available glass ionomer cements (GIC) at 5, 10, and 20 wt.%. The surface microhardness, acid resistance, and compressive strength were slightly reduced as the Nb_2_O_5_ concentration increased [106]. On the contrary, the radiopacity values of the Nb_2_O_5_ adhesives were significantly improved. It was concluded that the 5 wt.% concentration of Nb_2_O_5_ was associated with acceptable mechanical properties and a slightly elevated radiopacity. Thus, it seems that Nb_2_O_5_ in GIC had an inferior performance than Nb_2_O_5_ in resin-base materials.

### 4.8. Zirconium Oxide (ZrO_2_)

Zirconium oxide (ZrO_2_) has antibacterial effects against several bacterial and fungal species [107,108]. It can also mimic the natural appearance of the tooth structure [109]. Thus, multiple commercially available resin composite products contain ZrO_2_ in their formulations to enhance dental restorations’ optical and esthetic properties [71]. ZrO_2_ was also found to improve the µ-tensile bond strength when added at 5, 10, 15, and 20 wt.% to a dental primer and adhesive. For example, when it was added to a dental primer at 20 wt.%, the highest µ-tensile bond strength of 41.1 ± 12.9 MPa was achieved, significantly higher than the control (25.1 ± 10.9 MPa) [72]. 

## 5. The Use of Nanotubes in Restorative Dentistry 

Nanotubes have been used in the health area, especially for controlled drug release [110,111]. It is possible to use the chemical structure of nanotubes to chemically bond drugs on their surface, but they are more frequently used to carry molecules in their lumen. In the area of restorative dental materials, nanotubes have been used purely to confer bioactivity or improve physicochemical properties or carry antimicrobial molecules. Table 2 summarizes the different applications of nanotubes in restorative dentistry. Details about these purposes are described below.

### 5.1. Titanium Dioxide Nanotubes

Titanium dioxide (TiO_2_) has bioactivity, which was observed when this oxide was incorporated into polymers and stored in a simulated body fluid. In addition, calcium phosphates were observed on the surface of the polymer after storage [119]. TiO_2_ can be prepared as nanoparticles or nanotubes (Figure 4). Titanium dioxide nanotubes (nt-TiO_2_) provide osseointegration [120], inducing a proper osteoinduction and differentiation of pulp mesenchymal stem cells and adipose tissue in osteoblasts [121]. Moreover, nt-TiO_2_ already induced cell proliferation, alkaline phosphatase activity, and the expression of osteogenic proteins in an animal model [122].

Nanoparticles of TiO_2_ are the nanostructures of titania more tested in dental materials so far. In restorative materials, nanoparticles of TiO_2_ improved the hardness, flexural strength, degree of conversion, elastic modulus, shear bond strength, and antibacterial activity [123,124,125,126,127]. Currently, nt-TiO_2_ have also been tested because they have the advantage of presenting a hollow structure able to carry drugs, acting as a nanocarrier.

Nanocarriers are suitable for controlling the drug release and delivering it to specific sites [128,129]. nt-TiO_2_ have been shown to be an exciting platform to deliver therapeutic agents [128,130]. In a recent study, nt-TiO_2_ were added into a dental adhesive. These fillers carried a triazine-based molecule called 1,3,5-Trimethylhexahydro-1,3,5-triazine [126]. The nt-TiO_2_ loaded with the triazine-based molecule assisted in maintaining the µ-tensile bond strength over time and provided antibacterial activity. From future perspectives, this platform may be helpful to convey bioactivity for the adhesive.

### 5.2. Halloysite Nanotubes

Aluminosilicate nanoclays (Al_2_Si_2_O_5_(OH)_4_·*n*H_2_O) can be naturally found. Nanotubes of aluminosilicate nanoclays (halloysite nanotubes, HNTs) were already tested as filler for dental resins [112,113,114,115]. This material has a tubular structure, its outer surface is comprised of Si-O and silanol (SiH_4_O) groups, and the inner surface is composed of aluminol groups (Al-OH) [131]. The presence of silicon on the outer surface provides bioactivity for HNT. Therefore, the use of HNTs in methacrylate resins improved the biological properties of the polymer. Micro-Raman and Scanning Electron Microscopy (SEM) with an Energy Dispersive X-Ray Analysis (EDX) showed the presence of calcium phosphates on the surface of the dental resins doped with HNT when the polymers were immersed in artificial saliva [112] after 28 days. This result was attributed to the silanol and hydroxyl groups on HNTs’ outer surface due to their ability to induce mineral nucleation when they contact supersaturated fluids [112]. The reasons for the bioactivity provided by HNTs are described by the following [132]:(1)The H^+^ exchange and modification of silica network, creating a layer highly doped with SiO_2_;(2)The layer of SiO_2_ stimulates the precipitation of calcium and phosphate ions from the artificial saliva or SBF;(3)Differences in electronegativity between HNTs and the environment, which is rich in calcium and magnesium cations, induce the mineral nucleation and formation of apatite-like particles on the surface of the polymer doped with HNTs.

As aforementioned for nt-TiO_2_, HNTs can also carry drugs. This nanoclay was used as a drug delivery system with triclosan [113], a quaternary ammonium compound [114], chlorhexidine [115], and doxycycline [116] in dental resins for restoration. These platforms induced antibacterial properties and bioactivity for the resins.

### 5.3. Boron Nitride Nanotubes

Boron nitride nanotubes (BNNTs) have been investigated in the biomedical field due to their bioactivity [133] and ability to improve materials’ physical properties due to their high elastic modulus and superhydrophobic property [134]. They present biocompatibility and can be used to carry organic molecules, such as proteins [135], on their surface.

BNNTs were already tested in restorative resin-based materials. First, BNNTs were incorporated into an experimental adhesive [117], being tested regarding their physicochemical properties. While the degree of conversion remained stable, there was an increase in the contact angle and a decrease in the adhesives’ surface free energy. Furthermore, the mechanical properties and the resistance against softening in solvent improved with BNNT addition. The polymers doped with this material showed calcium phosphates on their surface after immersion for seven days in a simulated body fluid.

Another study analyzed the µ-tensile bond strength of adhesives after six months of aging [118]. The incorporation of BNNTs up to 0.1 wt.% into the adhesive improved the stability of the dental interface after six months compared to the adhesive without BNNTs. The bioactivity of dental resins with BNNT was shown in another research, and the proper biocompatibility was already demonstrated against pulp fibroblasts [118] and keratinocytes [136].

## 6. Future Perspectives and Conclusions

Advanced nanotechnology and basic science strategies have opened the doors for resin-based materials to reach a high level of bioactivity and mechanical performance. While the current data in using metal oxides and nanotubes are promising, essential points need to be emphasized. First, most of the conducted studies implemented immediate testing when the synthesized materials were investigated. It is critical to monitor the long-term performance of such new materials via the use of thermomechanical or hydrolytic aging [137]. Materials with excellent performance upon immediate testing may experience a significant deterioration following aging [137]. Thus, applying different aging approaches to re-examine such materials is recommended to obtain comprehensive data before in vivo investigations.

Another aspect that should be fully investigated is the cytotoxicity of the metal oxides and nanotube compounds. While several nanoparticles were investigated in the literature, studies to illustrate their effects after exposure and to report their potential toxicity are few. In dental practice, there is no exposure to nanoparticles when managing unset materials. Dental practitioners are mainly exposed to nanoparticle dust produced by grinding and polishing [138]. The lungs are the prime target organ. A risk assessment has shown a very low risk from inhaling nanoparticles as dust for currently available materials. However, there is a lack of data for a long-time exposure of dental nanoparticles for dental personnel. Although their exposure has been researched for many decades, no reports have been found on increased lung diseases for dental personnel [138]. Future investigations should consider evaluating the immediate and long-term cytotoxicity of different metal oxide nanoparticles and nanotubes in restorative materials. In the past, the level of mercury in human fluids and organs was investigated in the dental literature to explore the potential risk of dental amalgam [139]. Similar approaches could be implemented to investigate the potential cytotoxicity of the discussed materials in this review. 

The absence of a full evaluation of the designed resin-based materials was noted in most of the conducted studies. While the authors covered some aspects, other essential elements as well were not investigated. These aspects included the complete characterization and optimization of the material and other mechanical and physical properties that must be investigated. Moreover, the performance of the designed materials in different conditions, such as different pH environments, should be considered. More investigations are required to draw the full and optimum evaluation of such materials as well as to determine the potential metal oxide nanoparticles and nanotubes to be used in vivo. Another point to consider is the great difference between in vitro and in vivo conditions, especially in microbiology assays, where the bioactivity of materials is examined [140]. Therefore, using a clinical translational model to attempt these materials inside the oral cavity should be considered as an advanced strategy for a material’s evaluation. 

## Figures and Tables

**Figure 1 bioengineering-08-00146-f001:**
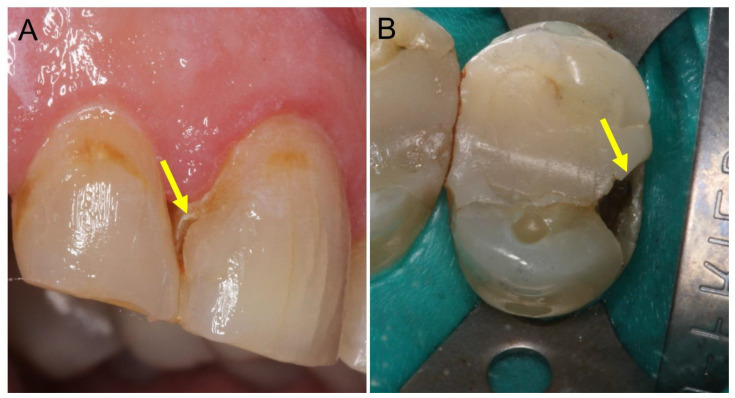
Clinical photos illustrate the two most common pathways for the failure of resin-based materials, secondary caries (**A**) and restoration fracture (**B**). (**A**) The arrow in the photo shows the lesion’s location at the tooth-restoration interface presented by yellow to brown discoloration. (**B**) The arrow in the photo shows the fracture location at the proximal wall of the tooth structure, most probably due to the high mechanical load induced by the masticatory force.

**Figure 2 bioengineering-08-00146-f002:**
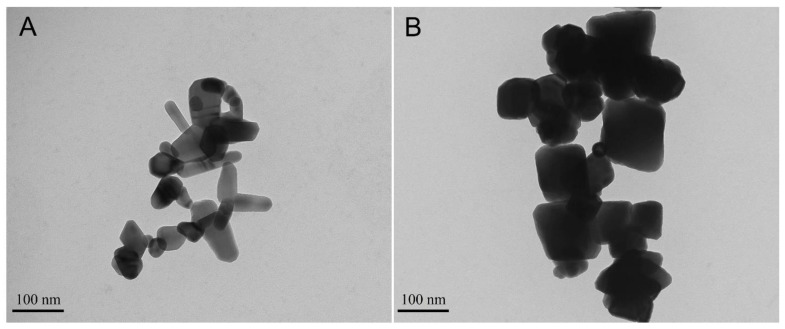
Transmission electron microscopy (TEM) image of oxide nanoparticles. Image (**A**) shows ZnO nanoparticles (Sigma-Aldrich Chemical Company, St. Louis, MI, USA) using 80.0 kV and 42,000× magnification. In the previous study, these ZnO nanoparticles were used as inorganic fillers to confer antibacterial activity for a dental adhesive. Image (**B**) shows Fe_3_O_4_ (Sigma-Aldrich Chemical Company, St. Louis, MI, USA) using 80.0 kV and 26,000× magnification. In a previous study, these ZnO nanoparticles were used as inorganic filler of a dental adhesive to improve the µ-tensile bond strength under simulated pulpal pressure and magnetic field application.

**Figure 3 bioengineering-08-00146-f003:**
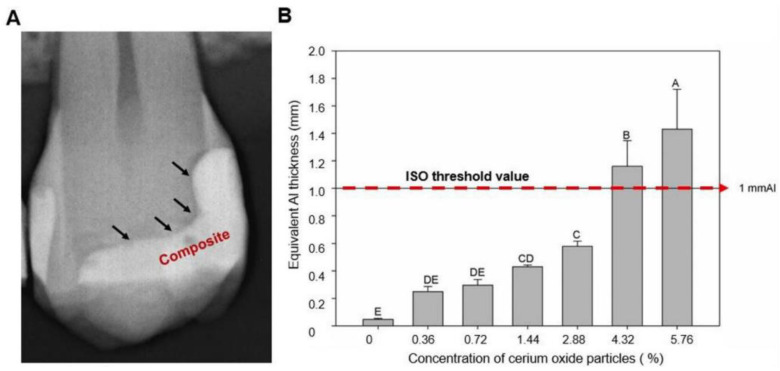
Image (**A**) illustrates a dental radiograph that shows a radiolucent line of interface between the tooth and the composite. The radiolucent area is the adhesive layer, formed by a dental adhesive without or with a low quantity of inorganic filler. Image (**B**) shows the radiopacity of a dental adhesive modified with different concentrations of cerium oxide particles. Different letters indicate statistical differences among groups (*p* < 0.05). Adapted from reference [65], with permission from Garcia et al., 2020.

**Figure 4 bioengineering-08-00146-f004:**
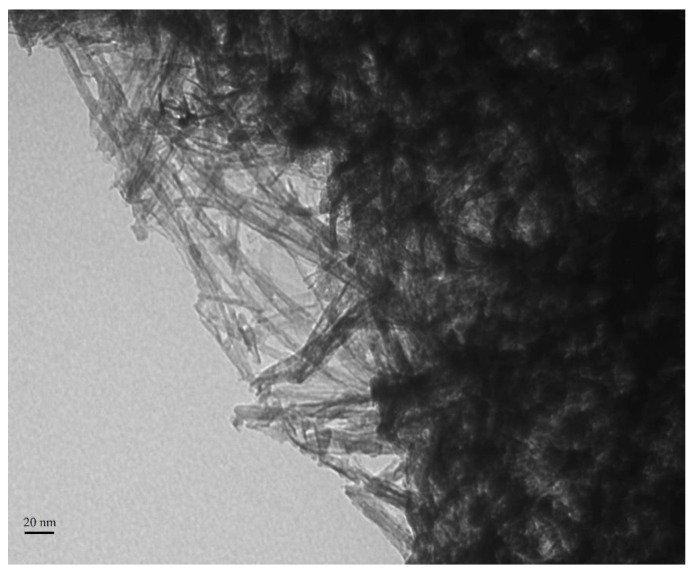
Transmission electron microscopy (TEM) image of titanium dioxide nanotubes. The image was acquired using 80.0 kV and 67,000× magnification. It is possible to observe the lumen of the elongated structures. These nanotubes were used purely and carried an antibacterial molecule in a dental adhesive in a previous study [126].

**Table 1 bioengineering-08-00146-t001:** The list and potential applications of metallic oxide compounds used in restorative dentistry.

Metallic Oxide	Potential Applications
Titanium Dioxide (TiO_2_)	- TiO_2_ nanoparticles improved the antibacterial properties against caries-related pathogens [54] - TiO_2_ enhanced the hardness and strength of resin-based materials [55] - TiO_2_ enhances the polymerization kinetics [56] and photo-catalytic activities [57]
Zinc Oxide (ZnO)	- ZnO nanoparticles improved the antibacterial properties against single [58] and multispecies biofilms [59] - Tetrapod-like zinc oxide whisker improved the mechanical properties of the designed resin composite [60]
Copper Oxide (CuO)	- Copper(I)-catalyzed azide-alkyne cycloaddition (CuAAC)-based resins reduced the growth of *S. mutans* biofilms [61] and improved the mechanical and polymerization properties [62]
Iron Oxide (Fe_2_O_3_)	- Fe_2_O_3_ improved the µ-tensile [63] and shear bond strength [64] of a dental adhesive
Cerium Oxide (CeO_2_)	- CeO_2_ significantly improved the radiopacity of dental adhesive [65], which can facilitate the differentiation between secondary caries and the adhesive layer
Tantalum Oxide (Ta_2_O_5_)	- Ta_2_O_5_ improved the fracture toughness of the designed materials [66,67] - Ta_2_O_5_ improved the radiopacity of a dental adhesive [68] - Ta_2_O_5_ quantum dots in dental adhesive inhibited the growth of *S. mutans* biofilm [69]
Niobium Pentoxide (Nb_2_O_5_)	- Nb_2_O_5_ improved the radiopacity of a dental adhesive [70]
Zirconium Oxide (ZrO_2_)	- ZrO_2_ can enhance the optical properties of resin composite restorations [71] - ZrO_2_ improved µ-tensile bond strength of dental adhesive [72]

**Table 2 bioengineering-08-00146-t002:** The list and potential applications of nanotubes materials used in restorative dentistry.

Nanotubes	Potential Applications
Titanium dioxide nanotubes (nt-TiO_2_)	nt-TiO_2_ were found effective in carrying 1,3,5-Trimethylhexahydro-1,3,5-triazine in a dental adhesive to exert antibacterial properties and improve the µ-tensile bond strength
Halloysite nanotubes (HNT)	HNT may deliver bioactive components around the incorporated materials by depositing calcium phosphate compounds, which may contribute to remineralization [112]. HNT can act as a drug delivery system with triclosan [113], quaternary ammonium compound [114], chlorhexidine [115], and doxycycline [116] in different resin-based materials
Boron nitride nanotubes (BNNTs)	BNNTs were found to increase the contact angle, decrease the surface energy, improve the mechanical properties [117], and preserved the bonding interface in a dental adhesive [118]

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
