# Peer review of "Metal Oxide Nanoparticles and Nanotubes: Ultrasmall Nanostructures to Engineer Antibacterial and Improved Dental Adhesives and Composites"

_bioengineering, 2021, doi:10.3390/bioengineering8100146_

Round 1
Reviewer 1 Report
Dear authors,
Congratulations for such an interesting and exhaustive review of MOx NPs and NTs within the field of dentistry. Last year we had the chance to count on a review of metallic NPs in dental biomaterials, but the publication your prepared was lacking.
I could only find a few minor issues which I would quickly present in order to improve the manuscript, from my humble point of view.
First of all, in the second section, there is a paragraph introducing antimicrobial activity and bioactivity (“In general, composite resins support…”). But suddenly, the next paragraph starts discussing the adhesive systems… And the reader is forced to wait 4 paragraphs more until finding the core information related to bioactivity. This was a bit strange for me. Maybe the text could be reorganized in order to preserve the logical presentation of ideas.
Besides this, in the forth section, it is said “the high density of these metal oxides provides a wide range of desired…”. But this is not exactly true. It is not the high density of the material per se, but the high density of corners and edges within the surface of MOx NPs, and their limited size. If you read carefully your own reference, you will see it says “The unique chemical and physical properties of MONPs are attributed to the high density and limited size of corners and edges on their surface.”. Therefore, I suggest clarifying this fact within the text.
In the 4.1 section, it says “The Rutile form of TiO2… cytotoxic properties and photo-catalytic [77]”. I suppose it refers to “photo-catalytic activity”.
Finally, in the section 5.5 it is said “In addition, tantalum demonstrates the capability to deposit appetite…”. I think the authors wanted to say “apatite”.
If the authors take care of this minor issues, I would recommend publishing the article.
Author Response
Comment #1: Congratulations for such an interesting and exhaustive review of MOx NPs and NTs within the field of dentistry. Last year we had the chance to count on a review of metallic NPs in dental biomaterials, but the publication your prepared was lacking. I could only find a few minor issues which I would quickly present in order to improve the manuscript, from my humble point of view.
Our response: We would like to thank the reviewer for your time and commitment to revising our paper. We believe that the provided comments have enhanced our manuscript’s quality, and we would like to express our thanks for the valuable comments and suggestions.
Comment #2: First of all, in the second section, there is a paragraph introducing antimicrobial activity and bioactivity (“In general, composite resins support…”). But suddenly, the next paragraph starts discussing the adhesive systems… And the reader is forced to wait 4 paragraphs more until finding the core information related to bioactivity. This was a bit strange for me. Maybe the text could be reorganized in order to preserve the logical presentation of ideas.
Our response: Thank you so much for this important comment. We have re-organized the paragraphs of that section to preserve the logical presentation of the paper. The new appearance of that part is now as:
“In general, composite resins and dental adhesives support a long-lasting and reliable treatment, the leading choice for anterior and posterior teeth restoration. However, factors related to patients and dentists influence the outcome. Therefore, the development of restorative materials with improved therapeutic properties, such as antimicrobial activity and bioactivity, has been researched. Thus, it is expected that these materials will help maintain treatments, especially for patients in a risk group.
Several approaches were followed to integrate bioactivity into resin-based restorative materials, including contact-killing and protein-repelling agents, inorganic metallic particles, and ion-releasing materials [3,4,21–23]. Contact-killing by Quaternary ammonium methacrylates (QAMs) were copolymerized in resins to accomplish antibacterial activities [13,24,25]. With the increase of alkyl chain length (CL) in its chemical structure, the antibacterial capability of QAMs could be increased [21]. Salivary proteins coating on the surface of the resin provides the medium for bacterial adhesion to the composite [26,27]. Thus, it is important to repel the salivary proteins from the composite surface. 2-methacryloyoxyethyl phosphorylcholine (MPC) is a common biopolymer with phospholipid polar groups [28,29]. Resins containing MPC had favorable protein-repellent properties, which enhanced the inhibition efficacy of the resins against cariogenic biofilms [30,31].
Release-based Antibacterial materials such as silver nanoparticles (Ag) show the ability to retain a long-lasting antimicrobial effect [32,33] and have less bacterial resistance than antibiotics [34]. Several studies had integrated nanoparticles of silver (Nag) into dental composites, which effectively inhibited the growth of Streptococcus mutans [35]. For remineralization purposes, bioactive composites enclose nanoparticles of amorphous calcium phosphate (NACP), which exhibited long-lasting calcium (Ca) and phosphate (P) ion releases [36–38]. The resins containing NACP have acid-neutralization capability to shield the tooth surfaces from acid challenges. In addition, the released ions stimulated tooth remineralization, resulting in the mineral regeneration in the tooth lesions [36–38]. It is essential to mention that some of these antibacterial agents showed limited long-term effects. In contrast, others showed satisfactory ability to reduce tooth demineralization, inhibit artificial secondary caries, and improve long-term bond strength. However, most of the available studies lack the clinical evaluation for those combined materials. Therefore, it is highly recommended to continue developing those antibacterial materials and investigate their performance clinically.
In addition to the fact that composite resins can be modified to provide bioactivity and antibacterial activity, adhesive systems also play an essential role in maintaining composite resin restorations. Therefore, these materials are also being modified. The most extensive study evaluating the clinical efficacy of adhesive systems was published in 2014. The study is a systematic review of papers in which adhesive systems were used in randomized clinical trials in non-carious cervical lesions from 1950 to 2013. The authors divided the adhesive systems into 3-step conventional, 2-step conventional, 2-step self-etching, and 1-step self-etching. The self-etching products were further divided according to pH into light and moderately strong, with pH ≥1.5; strong, with pH <1.5 [39]. The lowest annual failure rate was observed for the two-step self-etching adhesive system with moderate pH [2.5 (± 1.5)], followed by the conventional three-step adhesive system [3.1 (±2)] and the one-step self-etching adhesive system with moderate pH [3.6 (±4.3)], with no statistically significant difference between these systems. Higher annual failure rates were observed for strong 1-step self-etching adhesive systems [5.4 (4.8)], 2-step conventional adhesive system [5.8 (4.9)], and strong two-step self-etching adhesives [8.4 (7.9)], with no statistical difference between these three adhesive systems [39].
Regarding universal adhesive systems, the follow-up time is short compared to other systems. However, there are good results so far. The study with the most extended follow-up randomized clinical trial dates back to 2020 [40], and analyzed the survival of restorations in non-carious cervical lesions using universal adhesive and different acid etching strategies. After five years, retention/fracture rates were between 81.4-93%. The purpose of reporting these data is to highlight the importance of dental adhesives in improving the survival rate of resin composite restorations. As a result, several investigations were reported to modify dental adhesives to enhance their performance and impart bioactivity”.
Comment #3: Besides this, in the forth section, it is said “the high density of these metal oxides provides a wide range of desired…”. But this is not exactly true. It is not the high density of the material per se, but the high density of corners and edges within the surface of MOx NPs, and their limited size. If you read carefully your own reference, you will see it says “The unique chemical and physical properties of MONPs are attributed to the high density and limited size of corners and edges on their surface.”. Therefore, I suggest clarifying this fact within the text.
Our response: Thank you so much for this important comment. The sentence has been rephrased as:
“Besides, despite the size scale, the high density of corners and edges of these metal oxides provides a wide range of desired chemical and physical properties”.
Comment #4: In the 4.1 section, it says “The Rutile form of TiO2… cytotoxic properties and photo-catalytic [77]”. I suppose it refers to “photo-catalytic activity”.
Our response: Thank you so much. The word “activity” has been added.
Comment #5: Finally, in the section 5.5 it is said “In addition, tantalum demonstrates the capability to deposit appetite…”. I think the authors wanted to say “apatite”.
Our response: Thank you so much for this important observation. The word “apatite” has been placed instead of “appetite”.
We would like to thank the reviewer for the valuable comments and suggestions.
Reviewer 2 Report
Comments on the paper of Abdulrahman A. Balhaddad et al. entitled “Metal Oxide Nanoparticles & Nanotubes: Ultrasmall Nanostructures to Engineer Antibacterial and Improved Dental Adhesives and Composites”, Manuscript Number: 1420743
Comments and Suggestions for Authors:
The manuscript deals with the application of metal oxide nanoparticles & nanotubes incorporated into dental adhesives and composites, highlighting their antibacterial properties and applications in restorative dentistry.
The study appears to be carefully conducted with a sound analysis. The results and their interpretation are correct, and the paper is clearly written. I therefore generally recommend publication, however there are a couple of points that should first be addressed:
1) Was the influence of pH on the properties and quality of the new oxide materials investigated?
2) What was the stability of these materials and how their chemical or mechanical properties changed after long-term storage time?
3) How large was the spread among various TiO2 samples in terms of cytotoxic properties?
4) Are there any negative effects of the long-term exposure of these oxides to body fluids?
5) Some of the described materials exhibit toxic properties. Please explain if this was systematically investigated?
6) Which of the materials described by the authors is the most suitable for the usage in dentistry ?
7) Incorrect numbering of chapters: after chapter 4. 4, chapter 5. 4 appears. Please unify.
Author Response
Comment #1: The manuscript deals with the application of metal oxide nanoparticles & nanotubes incorporated into dental adhesives and composites, highlighting their antibacterial properties and applications in restorative dentistry.The study appears to be carefully conducted with a sound analysis. The results and their interpretation are correct, and the paper is clearly written. I therefore generally recommend publication, however there are a couple of points that should first be addressed.
Our response: We would like to thank the reviewer for your time and commitment to revising our paper. We believe that the provided comments have enhanced our manuscript’s quality, and we would like to express our thanks for the valuable comments and suggestions.
Comment #2: Was the influence of pH on the properties and quality of the new oxide materials investigated?
Our response: Thank you so much for this important comment. No reports was conducted to explore the performance of these materials in different pH. The following sentence has been added to address this point:
“While the authors covered some aspects, other essential elements as well were not investigated. These aspects include the complete characterization and optimization of the material and other mechanical and physical properties that must be investigated. Besides, the performance of the designed materials in different conditions, such as different pH environments, should be considered. More investigations are required to draw the full and optimum evaluation of such materials as well as determine the potential metal oxide nanoparticles and nanotubes to be used in vivo”.
Comment #3: What was the stability of these materials and how their chemical or mechanical properties changed after long-term storage time?
Our response: Thank you so much for this important comment. The following paragraph in the future perspectives has been added:
“Advanced nanotechnology and basic sciences strategies have opened the doors for resin-based materials to reach a high level of bioactivity and mechanical performance. While the current data in using metal oxides and nanotubes are promising, essential points need to be emphasized. First, most of the conducted studies implemented immediate testing when the synthesized materials were investigated. It is critical to monitor the long-term performance of such new materials via the use of thermomechanical or hydrolytic aging [137]. Materials with excellent performance upon immediate testing may experience a significant deterioration following aging [137]. Thus, applying different aging approaches to re-examine such materials is recommended to obtain comprehensive data before in vivo investigations”
Comment #4: How large was the spread among various TiO2 samples in terms of cytotoxic properties?
Our response: Thank you so much. The following paragraph has been added:
“Another aspect that should be fully investigated is the cytotoxicity of the metal oxides and nanotubes compounds. While several nanoparticles were investigated in the literature, studies to illustrate their effects after exposure and to report their potential toxicity are few. In dental practice, there is no exposure to nanoparticles when managing unset materials. Dental practitioners are mainly exposed to nanoparticles dust produced by grinding and polishing [138]. The lungs are the prime target organ. Risk assessment has shown a very low risk from inhaling nanoparticles as dust for currently available materials. However, there is a lack of data for a long-time exposure of dental nanoparticles for dental personnel. Although their exposure has been for many decades, no reports have been found on increased lung diseases for dental personnel [138]. Future investigations should consider evaluating the immediate and long-term cytotoxicity of different metal oxide nanoparticles and nanotubes in restorative materials. In the past, the level of mercury in human fluids and organs was investigated in the dental literature to explore the potential risk of dental amalgam [139]. Similar approaches could be implemented to investigate the potential cytotoxicity of the discussed materials in this review”
Comment #5: Are there any negative effects of the long-term exposure of these oxides to body fluids?
Our response: Thank you so much for this important comment. The following paragraph has been added:
“Another aspect that should be fully investigated is the cytotoxicity of the metal oxides and nanotubes compounds. While several nanoparticles were investigated in the literature, studies to illustrate their effects after exposure and to report their potential toxicity are few. In dental practice, there is no exposure to nanoparticles when managing unset materials. Dental practitioners are mainly exposed to nanoparticles dust produced by grinding and polishing [138]. The lungs are the prime target organ. Risk assessment has shown a very low risk from inhaling nanoparticles as dust for currently available materials. However, there is a lack of data for a long-time exposure of dental nanoparticles for dental personnel. Although their exposure has been for many decades, no reports have been found on increased lung diseases for dental personnel [138]. Future investigations should consider evaluating the immediate and long-term cytotoxicity of different metal oxide nanoparticles and nanotubes in restorative materials. In the past, the level of mercury in human fluids and organs was investigated in the dental literature to explore the potential risk of dental amalgam [139]. Similar approaches could be implemented to investigate the potential cytotoxicity of the discussed materials in this review”
Comment #6: Some of the described materials exhibit toxic properties. Please explain if this was systematically investigated?
Our response: Thank you so much for this critical comment. The following paragraph has been added:
“Another aspect that should be fully investigated is the cytotoxicity of the metal oxides and nanotubes compounds. While several nanoparticles were investigated in the literature, studies to illustrate their effects after exposure and to report their potential toxicity are few. In dental practice, there is no exposure to nanoparticles when managing unset materials. Dental practitioners are mainly exposed to nanoparticles dust produced by grinding and polishing [138]. The lungs are the prime target organ. Risk assessment has shown a very low risk from inhaling nanoparticles as dust for currently available materials. However, there is a lack of data for a long-time exposure of dental nanoparticles for dental personnel. Although their exposure has been for many decades, no reports have been found on increased lung diseases for dental personnel [138]. Future investigations should consider evaluating the immediate and long-term cytotoxicity of different metal oxide nanoparticles and nanotubes in restorative materials. In the past, the level of mercury in human fluids and organs was investigated in the dental literature to explore the potential risk of dental amalgam [139]. Similar approaches could be implemented to investigate the potential cytotoxicity of the discussed materials in this review”
Comment #7: Which of the materials described by the authors is the most suitable for the usage in dentistry ?
Our response: Thank you so much for this important comment. The following paragraph has been adjusted to address this point:
“The absence of a full evaluation of the designed resin-based materials was noted in most of the conducted studies. While the authors covered some aspects, other essential elements as well were not investigated. These aspects include the complete characterization and optimization of the material and other mechanical and physical properties that must be investigated. Besides, the performance of the designed materials in different conditions, such as different pH environments, should be considered. More investigations are required to draw the full and optimum evaluation of such materials as well as determine the potential metal oxide nanoparticles and nanotubes to be used in vivo. Another point to consider is the great difference between in vitro and in vivo conditions, especially in microbiology assays where the bioactivity of materials is examined [140]. Therefore, using a clinical translational model to attempt these materials inside the oral cavity should be considered as an advanced strategy for materials’ evaluation”
Comment #8: Incorrect numbering of chapters: after chapter 4. 4, chapter 5. 4 appears. Please unify.
Our response: Thank you so much. The chapters’ order has been adjusted and revised.
We would like to thank the reviewer for the valuable comments and suggestions.